# Virtual Therapy Complementary Prehabilitation of Women Diagnosed with Breast Cancer—A Pilot Study

**DOI:** 10.3390/ijerph20010722

**Published:** 2022-12-30

**Authors:** Oliver Czech, Katarzyna Siewierska, Aleksandra Krzywińska, Jakub Skórniak, Adam Maciejczyk, Rafał Matkowski, Joanna Szczepańska-Gieracha, Iwona Malicka

**Affiliations:** 1Department of Physiotherapy, Wroclaw University of Health and Sport Sciences, 51-612 Wroclaw, Poland; 2Lower Silesian Oncology, Pulmonology and Hematology Center, 53-413 Wroclaw, Poland; 3Department of Oncology, Wroclaw Medical University, 50-367 Wroclaw, Poland

**Keywords:** virtual reality, psychotherapy, physical health, quality of sleep, physical activity

## Abstract

Breast cancer is becoming an important issue due to its various consequences and epidemiology. Studies are showing that it extremely impacts the mental health as well as the physical activity of the patients. In addition to the most common symptom, which is fatigue, patients also have problems with the quality of sleep. Therefore, this study aimed to evaluate the effectiveness of virtual reality (VR) therapy in improving the mental state and quality of sleep, as well as increasing the physical activity (PA) of patients diagnosed with breast cancer. The study was conducted in a hospital’s Breast Unit and included patients at the time of diagnosis of malignant breast cancer. A total of 16 subjects randomly divided into experimental (*n* = 9), and control (*n* = 7) groups were measured with the Beck Depression Scale, Mental Adjustment to Cancer Scale, International Physical Activity Questionnaire, and Pittsburgh Sleep Quality Index at two timepoints. The experimental intervention consisted of a 2-week (8 sessions) Virtual Therapeutic Garden (VRTierOne) procedure performed daily for about 15 min. Significant differences were identified between groups in the interactions between the main factors seen in the destructive style of the Mini-Mac scale: F(1.14) = 4.82, *p* = 0.04, and between multiple experiments: F(1.14)= 5.54, *p* = 0.03 showing a significant reduction in the destructive style of coping with the disease in the study group after therapy (32.44 vs. 28.33, *p* = 0.003). The level of main effects [study] for the constructive style is F(1.14) = 3.93, *p* = 0.06 with a significant increase in constructive style in the study group (43.33 vs. 45.33, *p* = 0.044). Significant differences in levels of depression between multiple experiments: F(1.14) = 5.04, *p* = 0.04, show a significant reduction in the severity of depressive symptoms was found in the experimental group after therapy (13.33 vs. 8.11, *p* = 0.02). However, the analysis did not show significant differences between group analyses (*p* = 0.25). It seems that VR reduces the severity of depressive symptoms and reduces the destructive style and can be an effective option in improving the mental state of patients diagnosed with breast cancer.

## 1. Introduction

Neoplasms are a particularly important issue due to their health, psychological, and social consequences. Breast cancer is the most frequently diagnosed neoplastic disease in women. The peak incidence is between the ages of 65 and 69, but a significant number of cases are diagnosed in younger women [1]. Breast cancer more often affects women who are fully active in their professional, family, and social life.

Oncological diseases are a group of diseases that have an extremely strong impact on the mental health of patients. The diagnosis of cancer evokes strong emotional reactions in patients. Especially for women, breast cancer diagnosis causes a lot of fear, anxiety, and even depression. Researchers are defining several psychological comorbidities in breast cancer patients, e.g., half woman complex, depression, chronic stress, and fear. The prospect of long-term and burdensome treatments is often the cause anxiety, stress, and depression in patients. The prevalence of emotional distress in cancer patients ranges from 35% to 55%. The lack of control over treatment decisions reduces the patient’s quality of life [2,3,4]. Additionally, these problems may be aggravated by comorbid conditions. One of the most common cancer-related symptoms is fatigue, and its prevalence is estimated to be 60–90%. It is an important factor that can affect the overall quality of life and sleep. Quality of life also affects the level of PA [5,6,7]. A systematic review reported that home-based exercise interventions are effective in providing physiological and psychological benefits for cancer patients. Batalik et al. found that home-based exercise provides benefits in cardiorespiratory fitness, strength, PA level, Health-Related Quality of Life (HRQOL), and body composition. Thus, the investigation of similar approaches focusing on counteracting the negative physical and/or psychological side effects of cancer treatment is justified [8]. It also seems beneficial to implement the results from an overall cancer population and evaluate the effectiveness in a more specific cancer patients’ group [9]. Since breast cancer remains the most frequently diagnosed malignancy in women around the world, this population has been investigated in this pilot study. 

It has been proven that increasing the PA levels of patients has a beneficial effect on their mental health. Therefore, daily exercise and kinesiotherapy should be an integral part of oncological treatment. Research confirms that greater effectiveness can be obtained by using combination therapy [10,11]. Hence, it seems justified to use a therapy program that combines kinesiotherapy and psychotherapy. The variety of psychological symptoms and their causes qualify most cancer patients for psychotherapy, whether due to anxiety, depression, fear, or stress. Staff shortages, financial inefficiency, as well as the ever-growing number of patients are reasons to search for modern solutions that can meet the patient’s needs for rehabilitation. The maintenance-free and low-invasive nature of virtual reality is the reason that the use of VR in medical services is gaining more and more supporters and confirming feasibility in surgery, rehabilitation, dentistry, and neurology, among other fields. It has been shown to be effective in cognitive and motor functions improvement [12,13]. VR-based therapy for different mental health conditions have demonstrated promising results. VR seems to be an innovative and effective tool that can positively affect the quality of life and lead to the reduction of undesirable effects of oncological treatment, translating into the promotion of a more active lifestyle among cancer patients [14].

According to this evidence, it is assumed that the advantages of using VR can become an element of prehabilitation. Wu et al. [9] tried to investigate the feasibility of prehabilitation as part of the breast cancer treatment pathway. According to the results, 93% of the participants rated the prehabilitation as satisfactory. The prehabilitation program led to a reduction of anxiety scores in the prehabilitation group. Although the research was carried out on a small group of patients, similar conclusions were also obtained by scientists in other research centers. Brahmbhatt et al. [15] examined the feasibility of prehabilitation in breast cancer surgery patients. The results of this study suggest that prehabilitation affects not only the patients’ mental wellbeing but also can have an impact on physical indicators. The authors found a clinically significant increase in the 6-min walk distance from baseline to the preoperative assessment. Unfortunately, both cited studies were conducted on a small sample size. The poor number of publications and the small number of breast cancer patients examined for prehabilitation is also confirmed by a systematic review by Toohey et al. [16]. The publication included 14 studies from which most results showed that prehabilitation can improve outcomes, including physical function, quality of life, and psychosocial variables. The authors also indicate the need to improve future research in terms of sample size, methodological quality improvement, as well as follow-up measurement.

Analysis of the literature revealed that there are no studies investigating the impact of VR on daily PA and quality of sleep. Only results of psychometric assessments of participants were found. As PA is the basic intervention recommended in oncological patients, we decided to analyze whether VR therapy can affect the PA levels and sleep quality of studied patients. Therefore, our study aimed to evaluate the effectiveness of VR therapy in improving the mental state and quality of sleep, as well as increasing the PA of patients diagnosed with breast cancer.

## 2. Materials and Methods

The study was conducted in the Lower Silesian Oncology, Pulmonology, and Hematology Center (LSOPaH) in Wroclaw, Poland. All interventions were applied in the LSOPaH Breast Unit. Patients were included in the study at the time of diagnosis prior to the multidisciplinary team (MDT) meeting, which establishes and proposes the treatment strategy. Patients with cognitive impairments and those who had received psychiatric treatment in the past or during the study were excluded. All participants provided written consent to take part in the project. The study received approval from the Ethical Committee of the University School of Physical Education in Wroclaw, Poland (number 18/2019) and the approval of the Institutional Review Board of Lower Silesian Center of Oncology, Pulmonology, and Hematology in Wroclaw, Poland. Figure 1 presents the study flow (Figure 1).

Participants were assigned into two groups, with nine subjects in the experimental group and seven subjects in the passive control group. Randomization was based on a computer-generated list, considering the inclusion and exclusion criteria. The measures were repeated across two time points: pre-intervention and post-intervention for the experimental group and for the control group directly after diagnosis and 2 weeks after diagnosis (similar time points).

The experimental group received 2 weeks of Virtual Therapeutic Garden (VRTierOne) therapy sessions. The sessions lasted about 15 min and were performed every day. Each experimental-group patient had a total of eight therapeutic sessions. The sessions transferred patients to a virtual garden in which the therapist’s voice could be heard. The VR system consisted of VR goggles and two controllers (manipulators) plugged into a PC. By using a head-mounted display and the phenomenon of total immersion, VR therapy provides intense visual, auditory, and kinesthetic stimuli. The effects are calming and mood-improving. It also can help patients recognize their psychological resources and motivate them to take part in the rehabilitation process more actively. In the virtual garden, the symbols and metaphors used are based on the Ericksonian psychotherapeutic approach. The most important is the Garden of Revival, which symbolizes the patient’s health [17].

Standardized questionnaires were used to measure the studied parameters. The Beck Depression Scale (BDI) was used to assess the level of depression. The BDI is a relevant psychometric instrument showing high reliability, the capacity to discriminate between depressed and non-depressed subjects, and improved concurrent, content, and structural validity. The questionnaire consists of 21 questions, covering cognitive-affective symptoms and somatic symptoms that accompany mood disorders. A higher total score indicates more severe depressive symptoms [18]. The Mental Adjustment to Cancer Scale (Mini-MAC) was used to assess the participant’s response to the cancer diagnosis. The mini-MAC questionnaire is a widely used tool to assess coping strategies among cancer patients. The questionnaire assesses strategies used to cope with the disease. The scale consists of 29 items to which the patient answers according to a 4-point Likert scale: definitely not, rather not, rather yes, and definitely yes. The results range from 7 to 28 points. The scores are divided into four categories: anxious preoccupation, fighting spirit, helplessness-hopelessness, and positive redefinition [19]. A patient’s level of PA was measured using the International Physical Activity Questionnaire (IPAQ). IPAQ provides an estimate of PA and sedentary behavior for adults. It consists of seven questions reflecting on the activities of the previous 7 days according to five domains: (1) occupational PA; (2) transportation PA; (3) housework, house maintenance, and caring for family; (4) recreation, sport, and leisure-time PA; and (5) time spent sitting. IPAQ expresses PA in MET-min/week units, which allows respondents to be classified into one of three categories of activity: insufficient (less than 600), sufficient (600–1500 or 600–3000), or high (more than 1500 or 3000 MET-min/week) [20]. Additionally, participants’ sleep quality was tested using the Pittsburgh Sleep Quality Index (PSQI). The PSQI is an efficient indicator of the quality and pattern of sleep. The questionnaire consists of 19 individual items. It assesses sleep quality on seven categories—subjective sleep quality, sleep latency, sleep duration, habitual sleep efficiency, sleep disturbances, use of sleeping medication, and daytime dysfunction. Each item is scored on a 0–3 interval scale. The global index score is calculated by totaling the seven categories’ scores, providing an overall score ranging from 0 to 21, where a lower score denotes a healthier sleep quality [21].

Statistica 12 software (StatSoft, Cracow, Poland) was used to perform all statistical calculations and analyses. For continuous variables, the mean and standard deviations were calculated. Percentage points were used for categorical variables. The normality of the data was examined using Shapiro–Wilk’s test. Differences between groups for demographical data were examined using t-tests and Chi-square test for measurement data. One-way Repeated Measures Analyses of Variance (ANOVA) with a group as a categorical factor were applied. For statistically significant outcomes, post hoc tests (LSD) were used. The level of significance was set at an α value of <0.05.

## 3. Results

Participants were women with the diagnosis of a malignant breast tumor. Estrogen receptor expression was found in over 90% of patients in the study group, and the expression of progesterone receptors was seen in almost 70% of participants. HER2-positive breast cancer was found in more than 25% of the patients, and a Ki-67 proliferation index > 25% was observed in 56% of participants. According to cTNM cancer staging, 13% of the participant were at cTis (DCIS) stage. Moreover, 44% of patients were in stage cT1N0M0. The CT2N0M0 stage was assessed in 25% patients, and for stages cT2N1M0, cT3N0M0, and cT4N3M0, the percentage was 6% each. The baseline demographic characteristics of participants included in this study is presented in Table 1.

Significant differences were identified between groups in the interactions between the main factors seen in the destructive style of the Mini-Mac scale: F(1.14) = 4.82, *p* = 0.04, and between multiple experiments: F(1.14) = 5.54, *p* = 0.03 show a significant reduction in the destructive style of coping with the disease in the study group after therapy (32.44 vs. 28.33, *p* = 0.003). The level of main effects [study] for the constructive style is F(1.14) = 3.93, *p* = 0.06, with a significant increase in constructive style in the study group (43.33 vs. 45.33, *p* = 0.044).

Significant differences in levels of depression between multiple experiments: F(1.14) = 5.04, *p* = 0.04, showing that a significant reduction in the severity of depressive symptoms was found in the experimental group after therapy (13.33 vs. 8.11, *p* = 0.02,). However, the analysis did not show significant differences in between group analysis (*p* = 0.25).

In addition, the study group saw a maintenance in the quality of sleep level, with a significant deterioration of these parameters in the control group. There was also an increase in PA in experimental conditions compared to the control group with a moderate significance level.

All the obtained results are presented in detail in Table 2.

## 4. Discussion

This study was designed to assess the effectiveness of virtual therapy as a pre-rehabilitation intervention for women with breast cancer. Due to the debilitating nature of oncological treatments, it is extremely important to prepare patients for treatment. The pre-rehabilitation period focuses primarily on improving one’s physical condition and motivation to engage in PA. Developing a habit in patients allows them to maintain an appropriate level of PA during treatment. The phenomenon of immersion has enormous abilities to affect human consciousness, which, combined with therapy in the Ericksonian modality, creates a tool with great potential in the treatment of diseases characterized by a high mental burden. VR therapy is a promising technology to improve the quality of oncological rehabilitation. Nevertheless, more extensive research is needed in this area.

In the last decade, research using VR as a supplementary or alternative treatment has gained the interest of scientists. We noticed the growing popularity of VR in scientific research, as seen in the systematic review and meta-analysis carried out in 2021. However, in the included studies, VR was used as a distractor during chemotherapy. Although the use of the technology was slightly different, it is indisputable that most of the work using VR covers aspects of mental health. Most studies on the effects of VR on mental health show statistically significant results or trends in favor of VR groups [22]. However, the existing studies are insufficient to support the advantages of VR rehabilitation as a standalone intervention over conventional therapy. This may be a result of the small sample sizes and poor quality of the published papers. The results of this pilot study are promising. In our study, there was a significant reduction in depressive symptoms in the experimental group, and patients using VR coped with the diagnosis better. The results of our study are in line with previous research using the VR TierOne device. The same system has been used in chronic obstructive pulmonary disease (COPD) patients and has significantly reduced the severity of depressive symptoms and stress levels [23]. Similar results were obtained using VR TierOne in a group of coronary artery disease (CAD) patients [17] and older women with depressive symptoms [24]. In both studies, a control group of patients received Schulz Autogenic Training for comparison. Improvement in the psychophysical state after diagnosis is of great importance during treatment and rehabilitation. Mood and well-being have been shown to have a direct impact on the quality of therapy and the patient’s participation in the treatment process. Jóźwik et al. [25] also proved that VR intervention can improve the patient’s participation in treatment and improve the overall efficacy of rehabilitation. According to these results, VRTierOne is an effective tool to reduce patients’ stress levels.

During the ongoing COVID-19 pandemic, the therapeutic garden has also been used to improve the mood of hospitalized patients. The study design was a pragmatic pilot study with 40 patients. The intervention was a weekly self-administered at-home VR-based training protocol consisting of two parts: a 10-min VR video entitled “Secret Garden” and social exercises. The results indicated a significant reduction in stress and depression levels. This study is extremely important because it confirms that the effects of VR TierOne therapy last for at least 2 weeks, which was confirmed in a follow-up study [26].

According to actual evidence, VR can also be successful in improving sleep quality. Horesh et al. [27] investigated the effectiveness of VR on the overall wellbeing of ovarian and breast cancer patients. The authors assessed the impact on patients’ stress, distress, quality of life, illness perception, and sleep difficulties. The VR intervention has led to an improvement in the participants’ wellbeing but also was conducted on a small sample size. Chang et al. [28] examined whether VR can improve the quality of sleep of elderly patients with disabilities. Although the population differs from cancer patients, some of the symptoms overlap. Research confirms that VR can improve the quality of sleep despite a significant disease burden. The authors also found a reduction in the level of stress in these patients, which may indirectly affect the overall quality of life.

Researchers also agree that VR therapy can lead to increased physical activity in different populations. Evans et al. [29] investigated the energy expenditure of VR exergaming in healthy adults. Results from this experiment suggest that VR games can elicit varying degrees of PA intensity levels in young healthy adults. The games rated highest in enjoyment required mostly arm movement and a perceived light exertion. In a recent systematic review by Qian et al., physiological, psychological, and rehabilitative outcomes were measured during VR sessions [30]. Authors suggest that VR training has a positive impact on all the mentioned outcomes. They also concluded that the effects of VR therapy can be compared with conventional exercise. Unfortunately, this review also points to a gap in research, highlighting the small number of available publications.

However, VR TierOne is not the only system that is based on VR that can support patients in different clinical departments. There is research using the therapeutic garden environment to treat people with depressive symptoms. In 2013, Baños et al. proposed two virtual environments that simulated environments found in nature. The first aimed to improve the feelings of joy, whereas the second focused on relaxation in elderly participants. Both included a blue sky and a green field that the participants could explore. During the stroll, the voice instructed the user and told stories to evoke specific emotions. The study concluded that after using VR, there was a significant decrease in negative mood scores. The study used pre- and post-analysis during the two treatment sessions, using no control conditions and a touch screen as a VR medium [31]. This makes it difficult to relate our results to this study. Although the publications differ methodologically, the results emphasize the effectiveness of VR in improving psycho-social parameters. This dependence justifies the use of VR in clinical settings to improve the quality of treatment.

Due to the pilot study design, there were some limitations in our study. A sample size calculation was not performed, and a relatively small number of patients participated in the study. The small group sizes may increase the risk of bias. Moreover, a lack of follow-up assessments was a limitation, which makes it impossible to assess the durability of the obtained effect. The lack of blinding of the assessors could be considered a high-risk source of bias. However, the authors maintain that the obtaining of the results and their analysis were both done with scientific diligence and honesty. Despite these limitations, this research allows conclusions to be drawn and is another study supporting the legitimacy of implementing VR technology into modern therapy and treatment.

## 5. Conclusions

VR reduces the severity of depressive symptoms and reduces the destructive style and can be an effective option in improving the mental state of patients diagnosed with breast cancer.

## Figures and Tables

**Figure 1 ijerph-20-00722-f001:**
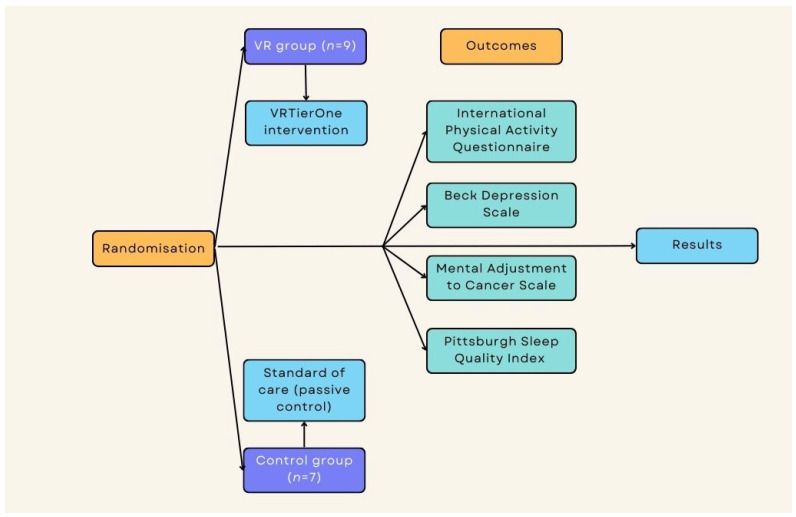
Study flow.

**Table 1 ijerph-20-00722-t001:** Baseline demographic characteristics.

Variable	Experimental	Control	*p*-Value *
*n*	9	7	-
Age, [years]; mean (SD)	50.59 (12.64)	59.55 (7.85)	0.12
BMI, [kg/m^2^]; mean (SD)	26.61 (5.77)	23.83 (2.41)	0.26
Education, *n* (%)			0.09
Primary/Vocational	3 (33)	3 (43)	
Secondary	0 (0)	3 (43)	
Incomplete Higher Education	1 (11)	0 (0)	
Higher Education	5 (56)	1 (14)	
Marital status, *n* (%)			0.49
Single	0 (0)	0 (0)	
Married	7 (78)	5 (71)	
Cohabiting	1 (11)	0 (0)	
Widowed	0 (0)	0 (0)	
Divorced	1 (11)	2 (29)	
Material status, *n* (%)			0.39
Bad	0 (0)	1 (14)	
Average	3 (33)	2 (29)	
Good	4 (45)	4 (57)	
Very Good	2 (22)	0 (0)	

Notes: SD—Standard Deviation, BMI—Body Mass Index, *—*t*-test or Chi-square test as appropriate.

**Table 2 ijerph-20-00722-t002:** Primary outcomes—the pre-post comparison of results as well as between groups comparison and the calculation of multiple experiments effect for all the measured outcomes.

Outcome	Experimental	Control	*p*-Value (Between Groups)
Mini Mental Adjustment to Cancer			
Destructive style			
Before	32.44 (6.75), (27.84: 37.42)	26.14 (5.98), (20.92: 31.35)	0.20
After	28.33 (8.31), (23.38: 33.27)	26.00 (4.44), (20.39: 31.60)	
*p*-value	**0.003**	0.917	
*p*-value Between multiple experiments	**0.033**		
Constructive style			
Before	43.33 (4.39), (40.37: 46.28)	44.29 (3.77), (40.93: 47.63)	0.886
After	45.33 (5.55), (41.93: 48.72)	45.00 (3.42), (41.14: 48.85)	
*p*-value	**0.044**	0.489	
*p*-value Between multiple experiments	0.067		
Beck Depression Inventory			
Before	13.33 (5.57), (8.86: 17.80)	9.00 (7.07), (3.92: 14.07)	0.31
After	8.11 (6.17), (3.89: 12.32)	7.00 (5.51), (2.21: 11.78)	
*p*-value	**0.028**	0.421	
*p*-value Between multiple experiments	**0.004**		
Pittsburgh Sleep Quality Index			
Before	7.22 (3.27), (4.94: 9.50)	3.14 (2.81), (0.55: 5.72)	0.088
After	6.22 (3.46), (4.05: 8.39)	5.71 (2.36), (3.25: 8.17)	
*p*-value	0.438	0.091	
*p*-value Between multiple experiments	0.420		
International Physical Activity Questionnaire (expressed in MET)			
intense physical activity			
Before	26.67 (80.00), (−148.57: 201.90)	137.14 (362.85), (−61.56: 335.84)	0.390
After	142.22 (316.93), (−100.63: 385.07)	228.57 (367.85), (−46.79: 503.94)	
*p*-value	0.396	0.551	
*p*-value Between multiple experiments	0.317		
moderate physical activity			
Before	306.67 (420.48), (1.24: 612.08)	271.43 (436.02), (−74.88: 617.74)	0.06
After	1293.33 (1452.17), (493.64: 2093.02)	222.86 (328.11), (−683.90: 1129.61)	
*p*-value	**0.036**	0.912	
*p*-value Between multiple experiments	0.166		
walking			
Before	1001.00 (973.08), (321.97: 1680.02)	820.29 (917.78), (50.34: 1590.22)	0.841
After		1767.86 (3541.21), (−209.95: 3745.67)	
*p*-value	0.721	0.365	
*p*-value Between multiple experiments	0.362		
overall			
Before	1334.33 (116.97), (451.71: 2216.95)	1228.86 (1375.79), (228.06: 2229.65)	0.685
After	2761.06 (2010.99), (739.50: 4782.60)	2219.29 (3641.98), (−72.93: 4511.51)	
*p*-value	0.184	0.407	
*p*-value Between multiple experiments	0.140		

Notes: before and after variables are expressed as means (Standard Deviation) including 95% CI, bold indicates statistical significance.

## Data Availability

Data is available from the corresponding author upon reasonable request.

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
