# Peer review of "Virtual Therapy Complementary Prehabilitation of Women Diagnosed with Breast Cancer—A Pilot Study"

_ijerph, 2022, doi:10.3390/ijerph20010722_

Round 1

Reviewer 1 Report

The paper sounds interesting. However, there are some areas that need to be clarified and revised.

Introduction

-      The authors wrote mainly about general oncological diseases rather than breast cancer. The recent systematic review (Ref. 8) is also not specific to breast cancer. (lines 38-69) I recommend the author to rewrite the introduction part to be more specific to breast cancer, which will narrow the scope and be more specific. The topic is about the prehabilitation but there was no any information about the benefit of prehabilitation especially in breast cancer survivors.

Research design

-      Why do the authors did the pilot study, not the RCT with assessor blind?

-      Why did not the author calculate the sample size?

-      How were the participants recruited in this study?

-      Which staging was the CA breast?

-      How did the author evaluate cognitive impairment?

-      How did the authors randomize the participants?

-      These statements are result “Participants were women with the diagnosis of a malignant breast tumor. Estrogen receptor expression was found in over 90% of patients in the study group, and the expression of progesterone receptors was seen in almost 70% of participants. HER2-positive breast cancer was found in more than 25% of the patients and a Ki-67 proliferation index > 25% was observed in 56% of participants.” I recommend authors to put them in the result.

-      The experimental group received 2 weeks of Virtual Therapeutic Garden

(VRTierOne) therapy sessions for 15 minutes per session per day for 8 sessions. Why did the authors use this protocol and was the duration of VR enough?

-      What kind of activity did the control group receive?

-      How to interpret the Mental Adjustment to Cancer Scale (Mini-MAC), which was used to assess the participant’s response to cancer diagnosis?

Results

-      Authors haven’t shown the study diagram.

-      Demographic data for both groups as well as all standardized questionnaires used to measure the parameters studied should be analyzed and demonstrated both within and between group comparison including 95% CI.

-      Both figures should have any symbol that shows a significant difference.

Discussion

-        Many studies demonstrated the effect of VR in depressive and anxiety reduction.  VR TierOne device is commonly used in stress and anxiety reduction. The authors should discuss how the VR TierOne device can improve physical activity and sleep quality? Are there any evidence supports?

Conclusion

-      The authors concluded that VR can also contribute to their adoption of an active attitude throughout the treatment and rehabilitation process. I think it is not relevant because this study has not shown any result to demonstrate the adoption to the treatment and rehabilitation process.

Author Response

Dear Reviewer,

Thank you for your deep and knowledgeable revision of this manuscript. The following are our answers:

  1. The authors wrote mainly about general oncological diseases rather than breast cancer. The recent systematic review (Ref. 8) is also not specific to breast cancer. (lines 38-69) I recommend the author to rewrite the introduction part to be more specific to breast cancer, which will narrow the scope and be more specific. The topic is about the prehabilitation but there was no any information about the benefit of prehabilitation especially in breast cancer survivors.
  • Thank you for pointing this out. We have wrote about general oncological diseases, due the information is also accurate for breast cancer patients. However, we have added some information about breast cancer specifically. Also prehabilitation has been described according to several citations. The introduction should be more consistent now.
  1. Why do the authors did the pilot study, not the RCT with assessor blind?
  • The research was carried out during the covid 19 pandemic with limited access and a full sanitary regime. The recruitment of patients was not smooth. Obtaining positive results prompted the authors to prepare the work as a pilot study. Currently, main studies based on RCTs have been prepared, however, we decided that the results obtained from the pilot study should be presented to a wider group of scientists.
  1. Why did not the author calculate the sample size?
  • Due to limited access to the study group. Further studies will take into account the sample size, we included the lack of this calculation in the limitations of the study
  1. How were the participants recruited in this study?
  • Patients are referred to researchers by treatment process coordinators, who are also responsible for bringing patients, e.g. for additional tests and consultations in the day-to-day operation of the hospital
  1. Which staging was the CA breast?
  • Thank you for the suggestion. The stage percentage has been added into the results description.

  1. How did the author evaluate cognitive impairment?
  • Only on the basis of a medical card containing the results of a medical examination

  1. How did the authors randomize the participants?
  • Randomization was based on a computer-generated list, considering the inclusion and exclusion criteria. This information ha salso been added to the manuscript.
  1. These statements are result “Participants were women with the diagnosis of a malignant breast tumor. Estrogen receptor expression was found in over 90% of patients in the study group, and the expression of progesterone receptors was seen in almost 70% of participants. HER2-positive breast cancer was found in more than 25% of the patients and a Ki-67 proliferation index > 25% was observed in 56% of participants.” I recommend authors to put them in the result.
  • We agree with the reviewer's opinion, we have transferred this data to the results.
  1. The experimental group received 2 weeks of Virtual Therapeutic Garden (VRTierOne) therapy sessions for 15 minutes per session per day for 8 sessions. Why did the authors use this protocol and was the duration of VR enough?
  • This methodology is suggested by the inventor, as well as studies on other wards (pulmonology, cardiology, neurology). We have used the suggested settings.

  1. What kind of activity did the control group receive?
  • Control conditions was meant as standard of care procedures, in this case we have passive control conditions.

  1. How to interpret the Mental Adjustment to Cancer Scale (Mini-MAC), which was used to assess the participant’s response to cancer diagnosis?
  • The Mini-MAC scale is a tool dedicated for cancer patients. It is a self-descriptive tool. The respondent on his/her own evaluates, using a four-point scale, to what extent a given statement applies to him/her at present. According to the method, there are two styles of coping with disease: constructive (strategies: fighting spirit and positive redefinition), destructive (strategies: helplessness-hopelessness and anxious preoccupation). The Mini Mac scale is a standardized measurement tool

  1. Authors haven’t shown the study diagram.
  • Thank you for pointing this out. We have updated the manuscript with the study flow.

  1. Demographic data for both groups as well as all standardized questionnaires used to measure the parameters studied should be analyzed and demonstrated both within and between group comparison including 95% CI.
  • Thank you for this comment. The results section has been improved and more precise data has been uploaded.

  1. Both figures should have any symbol that shows a significant difference.
  • Thank you for the suggestion, we decided to replace the figures with tables to present all the necessary data more transparent.

  1. Many studies demonstrated the effect of VR in depressive and anxiety reduction. VR TierOne device is commonly used in stress and anxiety reduction. The authors should discuss how the VR TierOne device can improve physical activity and sleep quality? Are there any evidence supports?
  • Thank you for pointing this out. We have updated the manuscript with several citations according to this comment. We believe it is justified to look for beneficial effects of VR on PA and quality of sleep.

  1. The authors concluded that VR can also contribute to their adoption of an active attitude throughout the treatment and rehabilitation process. I think it is not relevant because this study has not shown any result to demonstrate the adoption to the treatment and rehabilitation process.
  • Thank you for this comment. This statement came from already confirmed knowledge – positive mindset allows the patient to participate better in the rehabilitation process. Of course we have decided to skip this conclusion, and leave this thought to be extended in the next publication.

We hope the revised version is now suitable for publication.

Thank you again for the review.

Yours faithfully, Authors

Reviewer 2 Report

Title: Virtual therapy complementary prehabilitation of women diagnosed with breast cancer – a pilot study.

This study aimed to evaluate the effectiveness of Virtyal Reality therapy in improving the mental steate and quality of sleep, as well as increasing the physical activity of patients diagnosed with breast cancer.

Main comments

In general, the manuscript is well-written, maybe a little short, particularly the results. No tables in the whole document. Some specific comments are presented below.

0. Abstract

- PA and VR abbreviations. The first time appeared is mandatory to write the word with the abbreviation between brackets. Please, write in Line 14 “…physical activity (PA)...” and in Line 15 “…virtual reality (VR)…”. After these lines, only use PA or VR.

- Line 17: Doble point. Please write “…with breast cancer. The study…”.

1. Introduction

- Please include more articles and clinical trials with succesful results of VR in cancer diagnosis.

2. Materials and Methods

After line 91 the presence of a “study flow chart/diagram” figure could be visual to an easily understanding of this part.

3.Results

As a pilot study it could be recommended to expand the information in this point.

Absence of demographic table or patients characteristics table.

Figure 2 has poor quality, please add another of better quality.

4.Discussion

No comments.

5. Conclusions

No comments.

6. References

- Please review that all the references are in the same format.

- Please refer the journals with abbreviations as you did in “Int J Radiat Biol” (Line 279) or “J Clin Med” (Line 309).

- Line 284. More data after the Journal.

- Lines 273, 278, 288, 291, 295, 297, 300, 303, 305, 308, 311, 314 and 317: Year of publication after authors.

Author Response

Dear Reviewer,

Thank you for your deep and knowledgeable revision of this manuscript. The following are our answers:

  1. PA and VR abbreviations. The first time appeared is mandatory to write the word with the abbreviation between brackets. Please, write in Line 14 “…physical activity (PA)...” and in Line 15 “…virtual reality (VR)…”. After these lines, only use PA or VR.
  • Thank you for this comment. We have improved the text according to this suggestion.
  1. Line 17: Doble point. Please write “…with breast cancer. The study…”.
  • Thank you for this comment. The manuscript has been scanned for similar errors.
  1. Please include more articles and clinical trials with succesful results of VR in cancer diagnosis.
  • Thank you for your comment. The introduction section has been updated.
  1. After line 91 the presence of a “study flow chart/diagram” figure could be visual to an easily understanding of this part.
  • Thank you for this suggestion. The study flow diagram has been added to the manuscript.
  1. As a pilot study it could be recommended to expand the information in this point.
  • Thank your for the comment. The results section has been improved according to your suggestions.
  1. Absence of demographic table or patients characteristics table.
  • Thank you for pointing this out. The tables has been updated in the manuscript.
  1. Figure 2 has poor quality, please add another of better quality.
  • Thank you for the information, tables have been added for better clarity.
  1. Please review that all the references are in the same format.
  • Thank you for pointing this out. The references has been revised.
  1. Please refer the journals with abbreviations as you did in “Int J Radiat Biol” (Line 279) or “J Clin Med” (Line 309).
  • Thank you for pointing this out. The references has been revised.
  1. Line 284. More data after the Journal.
  • Thank you for pointing this out. The references has been revised.
  1. Lines 273, 278, 288, 291, 295, 297, 300, 303, 305, 308, 311, 314 and 317: Year of publication after authors.
  • Thank you for pointing this out. The references has been revised.

We hope the revised version is now suitable for publication.

Thank you again for the review.

Yours faithfully, Authors